# Iron Deficiency in Celiac Disease: Prevalence, Health Impact, and Clinical Management

**DOI:** 10.3390/nu13103437

**Published:** 2021-09-28

**Authors:** Miguel A. Montoro-Huguet, Santos Santolaria-Piedrafita, Pablo Cañamares-Orbis, José Antonio García-Erce

**Affiliations:** 1Department of Medicine, Faculty of Health Sciences, University of Zaragoza, 22002 Huesca, Spain; 2Gastroenterology, Hepatology and Nutrition Unit, St George’s Hospital, 20004 Huesca, Spain; ssantolariap@gmail.com (S.S.-P.); pablocanamaresorbis@gmail.com (P.C.-O.); 3Aragón Health Research Institute (IACS), Aragon Institute for Health Research (IIS Aragon), 50009 Zaragoza, Spain; 4Blood and Tissue Bank of Navarra, Pamplona, 31008 Navarra, Spain; jagarciaerce@gmail.com

**Keywords:** iron deficiency, iron deficiency anemia, celiac disease, malabsorption, micronutrient deficiencies, gluten-free diet, iron oral, iron intravenous, patient-blood management (PBM)

## Abstract

Iron is an essential nutrient to life and is required for erythropoiesis, oxidative, metabolism, and enzymatic activities. It is a cofactor for mitochondrial respiratory chain enzymes, the citric acid cycle, and DNA synthesis, and it promotes the growth of immune system cells. Thus, iron deficiency (ID) leads to deleterious effects on the overall health of individuals, causing significant morbidity. Iron deficiency anemia (IDA) is the most recognized type of anemia in patients with celiac disease (CD) and may be present in over half of patients at the time of diagnosis. Folate and vitamin B12 malabsorption, nutritional deficiencies, inflammation, blood loss, development of refractory CD, and concomitant *Heliobacter pylori* infection are other causes of anemia in such patients. The decision to replenish iron stores and the route of administration (oral or intravenous) are controversial due, in part, to questions surrounding the optimal formulation and route of administration. This paper provides an algorithm based on the severity of symptoms; its impact on the health-related quality of life (HRQL); the tolerance and efficiency of oral iron; and other factors that predict a poor response to oral iron, such as the severity of histological damage, poor adherence to GFD, and blood loss due to mucosal lesions.

## 1. Introduction

Iron is an essential nutrient to life, and its role in biology is enormous [1,2,3]. In fact, iron is required for erythropoiesis, oxidative, metabolism, and enzymatic activities, and it is a cofactor for mitochondrial respiratory chain enzymes, the citric acid cycle, and DNA synthesis [4]. It also promotes the growth of immune system cells. Iron deficiency (ID) is the most common deficiency state in the world, affecting more than two billion people globally. Although it is particularly prevalent in less-developed countries, it remains a significant problem in the developed world, where other forms of malnutrition have been almost eliminated [5]. Celiac disease (CD) is a well-recognized cause of IDA, even in asymptomatic patients, and, therefore, it must be considered in the differential diagnosis of IDA [6]. The prevalence of CD among people with anemia (and vice versa), its clinical consequences, and its management in specific contexts is discussed here, providing healthcare professionals with practical guidance and algorithms.

## 2. Iron Metabolism

Iron is an essential micronutrient with well-established contributions to body functions, such as the formation of red blood cells and hemoglobin, oxygen transport, cell division, energy metabolism, immunity, and cognition [7].

The body and cells need a very precise amount of iron: too much can be toxic and too little is bad for the metabolism [7]. Due to this toxicity, and in the absence of active excretion mechanisms, intestinal iron absorption is extremely limited and regulated tightly (barely 1–2 mg/day) to compensate for natural losses, and it is fundamentally based on recycling and a very precise circular economy. Therefore, internal turnover of iron is essential to satisfy the requirements of erythropoiesis (20–30 mg/d) [8].

ID refers to reduced iron stores and can progress to IDA, which is a more serious condition in which low iron store levels are associated with anemia. It is important to note that ID and anemia are not synonymous; a normal Hb level does not exclude ID (Figure 1). IDA can also be associated with other causes of anemia and other nutritional deficits.

The main cause of ID is an inadequate absorption or supply disorder due to an unbalanced diet that does not compensate for the increase in physiological needs at certain times in life (growth, pregnancy, or recovery) or after significant, acute, or chronic blood loss. Gastric and intestinal integrity, both organic and functional, is mandatory.

Dietary non-heme iron primarily exists in an oxidized (Fe^3+^) form that is not bioavailable and must first be reduced to the Fe^2+^ form by a ferrireductase enzyme, which uses vitamin C as a coenzyme, before being transported across the intestinal epithelium. This is accomplished by a carrier protein called divalent metal transporter 1 (DMT1) [9].

Nearly all absorption of dietary iron occurs in the duodenum. Several steps are involved, including the reduction of iron to a ferrous state, apical uptake, intracellular storage or transcellular trafficking, and basolateral release [8,9].

Once inside the intestinal epithelial cell, most Fe^2+^ is exported by ferroportin 1 across the basolateral membrane of the enterocyte (absorbed iron) and oxidized to Fe^3+^ by hephaestin before being bound by plasma transferrin. Ferroportin 1 is also expressed in hepatocytes, reticuloendothelial macrophages, and placental syncytiotrophoblasts [9].

Iron released into the circulation binds to transferrin and is transported to sites of use and storage. About 30–40% of the iron-binding capacity of transferrin is used in normal physiological conditions; thus, there is ∼4 mg of transferrin-bound iron, but this is the most important dynamic iron pool. Transferrin-bound iron enters target cells—mainly erythroid cells, but also immune and hepatic cells—through a highly specific process of receptor-mediated endocytosis [8,9].

In the cell, iron can be stored in two forms: in the cytosol as ferritin, and, after breakdown of ferritin, in the lysosomes as hemosiderin. Iron export from macrophages to transferrin is accomplished primarily by ferroportin 1, the same iron-export protein as expressed in the duodenal enterocyte [8,9]. The liver is the other main storage organ for iron, but RBC mass is the main storage “place”.

Iron homeostasis is regulated by two main mechanisms: an intracellular mechanism, which depends on the amount of iron available to the cell, and a systemic mechanism, in which hepcidin plays a crucial role [10].

Hepcidin is the main regulator of systemic iron homeostasis. Hepcidin coordinates the use and storage of iron. It is a mediator in the iron cycle between the liver and the intestine. Hepcidin acts by inhibiting intestinal iron absorption at the level of the basement membrane of the enterocyte, thereby inhibiting iron release by macrophages and enterocytes [10].

The positive regulation of hepcidin by the inflammatory response pathways to stress, its hepatic synthesis through (IL 6) interleukin 6, is an important critical event that triggers withdrawal and systemic iron sequestration due to its negative regulation of iron. Ferroportin causes an increase in iron levels, which, in turn, limit iron transport from the liver and macrophages to plasma; furthermore, it also inhibits the absorption of iron from the diet in the duodenum, which leads to iron restriction. Iron absorption in the presence of increased hepcidin is inhibited.

Anemia and hypoxia, as well as increased erythropoiesis, induce a cascade of changes that, individually or in combination, suppress hepcidin expression [10].

## 3. Laboratory Tests for the Detection of ID

ID is a progressive process in which iron stores fall from being replete to deplete and, finally, absent, consequently resulting in IDA. Progressive ID can be measured by a variety of biomarkers, as discussed below.

### 3.1. Full Blood Count, Blood Film and Red Cell Indices

A full blood count is performed routinely in patients with CD and may show low Hb, mean cell volume (MCV) (average volume (size) of the (RBC), mean cell hemoglobin (MCH) (average hemoglobin content in a RBC), and mean cell hemoglobin concentration (MCHC) (average hemoglobin concentration per RBC); a blood film may confirm the presence of microcytic hypochromic red cells. However, for milder cases of ID, the MCV may not have fallen below the normal range. Some analyzers will give a percentage of the hypochromic or microcytes red cells present. Both are fast and most sensitive markers of functional ID.

### 3.2. Serum Ferritin

Serum ferritin is a stable glycoprotein that accurately reflects iron stores in the absence of inflammatory change. As this glycoprotein is the first to become abnormal as iron stores decrease and since it is not affected by recent iron ingestion, it is examined in laboratory tests. The ferritin test is generally considered the best test to assess ID in patients with malabsorption, although it is an acute phase reactant, and levels will rise when there is active infection or inflammation. A serum ferritin < 30 g/L is a sensitive marker of ID, with <15 g/L being pathognomonic of ID with or without anemia. Ferritin concentrations below 50 or 100 g/L are highly suggestive of ID in the presence of inflammation or “mixed” anemia, whereas in the absence of inflammation, they reflect poor iron “reserves”.

### 3.3. Serum Iron (Fe) and Total Iron Binding Capacity (TIBC)

Serum Fe and TIBC are unreliable indicators of availability of iron to the tissue because of wide fluctuation in levels due to recent ingestion of Fe, diurnal rhythm, and other factors such as infection. However, in the presence of inflammation, transferrin saturation < 20% is a sensitive marker of ID.

### 3.4. Soluble Transferrin Receptor (sTfR)

Measurement of sTfR is reported to be a sensitive measure of tissue iron supply and is not an acute-phase reactant [11]. The transferrin receptor is a transmembrane protein that transports iron into the cell. Circulating concentrations of sTfR are proportional to cellular expression of the membrane-associated TfR and, therefore, give an accurate estimate of ID. There is little change in the early stages of iron store depletion, but once ID is established, the sTfR concentration increases in direct proportion to total transferrin receptor concentration. However, the higher cost of this test and the lack of standardization restricts its general availability [1,8,10].

### 3.5. Reticulocyte Hemoglobin Content and Reticulocytes

ID causes a reduction in reticulocyte number and reticulocyte hemoglobin concentration. However, anemia with a decreased (or inappropriately low) reticulocyte count may be due to deficiency of iron, vitamin B12, folate, or copper; medications that suppress the bone marrow; primary bone marrow disorders, including myelodysplastic syndrome (MDS), myelofibrosis, or leukemia, and very recent bleeding (within five to seven days before bone marrow compensation has occurred).

### 3.6. Red Cell Distribution (RDW)

Red cell distribution width (RDW) is a measure of the variation in RBC size, which is reflected in the degree of anisocytosis on the peripheral blood smear. RDW is calculated as the coefficient of variation (CV) of the red cell volume distribution (RDW = (standard deviation/MCV) × 100). A high RDW implies a large variation in RBC sizes, and a low RDW implies a more homogeneous population of RBCs. A high RDW can be seen in several anemias, including ID, vitamin B12 or folate deficiency, myelodysplastic syndrome (MDS), and hemoglobinopathies, as well as in patients with anemia who have received transfusions. In fact, it is a parameter for early detection of any erythropoiesis disorder. A review of the peripheral blood smear is often helpful in identifying the cause.

### 3.7. Bone Marrow Iron

A bone marrow sample stained for iron has been considered the gold standard for assessment of iron stores; however, this test is clearly too invasive and not practical for most patients with CD.

At the time of diagnosis, CD patients have some common nutritional deficiencies. These may include deficiencies of iron, vitamin B12, folate, and copper, which may occur in isolation or simultaneously. Some patients may have ID without anemia and must, therefore, be tested to discard ID before anemia will appear). Iron studies will identify ID (the most likely diagnosis for microcytic anemia). Mild microcytosis with iron studies showing low iron, low TIBC, and high-normal to high ferritin in the appropriate clinical context (e.g., chronic inflammatory condition with normal MCV prior to its development) is consistent with anemia of chronic disease (ACD). Figure 1 shows the evolution of the indicators of ID until the patient develops overt IDA.

Briefly, serum ferritin concentration < 30 g/L, transferrin saturation < 20%, and/or the presence of hypochromic microcytic erythrocytes (mean corpuscular volume < 80 fL, mean corpuscular Hb < 27 pg) are indicative of ID. In the presence of inflammation, transferrin saturation < 20% and ferritin > 100 g/L indicate a functional ID (iron sequestration).

## 4. Symptoms

Clinical symptoms and signs of IDA are usually nonspecific unless the anemia is severe. Some patients with IDA will be asymptomatic; others will have symptoms that may include the following:Symptoms of anemia, which may include weakness, headache, decreased exercise tolerance, fatigue, irritability, or depression. Asthenia, tiredness, and muscle weakness appear even without apparent anemia. IDA may also impair temperature regulation and may make one feel colder than normal.Neurodevelopmental delay (children).Lack of concentration and lower academic performance (adolescent).Worse physical performance (sport competition).Pica and pagophagia (ice craving).Beeturia (reddish urine after eating beets).Restless legs syndrome.

Because storage iron is depleted before a fall in Hb and iron is an essential element in all cells, symptoms of ID may occur even without anemia. These include fatigue, irritability, poor concentration, brittle nails, scratches, depapillation of the tongue, and hair loss.

Of particular interest are the symptoms that can occur in pregnancy [12,13,14,15,16,17,18,19,20,21], in children [22,23,24], and during the productive working age [25,26,27,28,29,30,31,32].

### 4.1. Pregnancy

ID may contribute to maternal morbidity through effects on immune function with increased susceptibility or severity of infections [13], poor work capacity and performance [14], and disturbances of postpartum cognition and emotions [15]. CD is frequently found in women of childbearing age. There is some evidence for the association between maternal ID and preterm delivery [16], low birth weight [17], possible placental abruption, and increased peripartum blood loss [18]. However, further research on the effect of ID, independent of confounding factors, is necessary to establish a clear causal relationship with pregnancy and fetal outcomes. The fetus is relatively protected from the effects of ID by upregulation of placental iron transport proteins [19], but evidence suggests that maternal iron depletion increases the risk of ID in the first three months of life by a variety of mechanisms [20,21].

### 4.2. Children

Impaired psychomotor and/or mental development are well described in infants with IDA and may also negatively contribute to infant and social emotional behavior [22]. They also have an association with adult onset diseases, although this is a controversial area [23,24]. Other common nutritional deficiencies in the patient with CD (folate, zinc, cobalamin, niacin, and biotin) contribute to the worsening of these symptoms.

### 4.3. Productive Working Age

ID and IDA are global health problems leading to deterioration in patients’ quality of life and more serious prognosis in patients with chronic diseases [25,26,27,28,29,30,31,32]. People with ID, even without anemia, in the productive working age may present some symptoms, such as loss of concentration and memory, foggy mind, asthenia, fatigue, and depressed mood, which can be very distressing, this being a frequent cause of presenteeism (decreased productivity at the workplace). In fact, there is evidence that HRQL in these patients is severely impaired in many dimensions (physical, social, and emotional). This is a point to bear in mind when considering the route of iron replenishment (oral versus intravenous) (see below).

### 4.4. Elderly

CD is also possible in the elderly, and, in fact, its incidence has increased in recent decades. A low index of suspicion by a physician may lead to diagnostic delay in recognition or to a distraction to other disorders. The following are three important aspects to consider: first, anemia may be present in up to 80% of cases, even in the presence of no digestive symptoms or in paucisymptomatic patients; second, some elderly patients will be seronegative, but this does not exclude the need to be investigated; and third, elderly patients are vulnerable to the effects of tissue hypoxia, and, in such cases, rapid and effective replenishment of iron stores is imperative when anemia is present (see below).

## 5. Prevalence of Celiac Disease (CD) in Patients with Anemia

### 5.1. Global Overview

CD is a chronic small intestinal immune-mediated enteropathy precipitated by exposure to dietary gluten in genetically predisposed individuals. Gluten is a complex of water-insoluble proteins from wheat, rye, and barley that are harmful to patients with CD, and, indeed, to confirm a diagnosis of CD (at least for adults), biopsies of the duodenum must be taken when patients are on a gluten-containing diet.

“Since the Oslo Consensus, published in 2013 [33], the scientific community has cknowledged different patterns of clinical presentation. «Classical» CD presents with signs and symptoms of malabsorption. Examples of classical CD are patients with diarrhea and steatorrhea, but also patients with weight loss and anemia or failure to thrive. In non-classical CD (before ‘atypical’ CD), the patient does not suffer from malabsorption (e.g., abdominal pain, diarrhea, or constipation, but without any evidence of malabsorption). This consideration is important because some patients with gastrointestinal symptoms with apparent functional criteria (e.g., irritable bowel syndrome) may ultimately be diagnosed with CD if a clinician with a high index of suspicion decides to investigate the cause of an ID (even without anemia) that any other cause cannot explain. Potential celiac disease (PCD) is defined by the presence of positive serum antibodies, HLA-DQ2/DQ8 haplotypes, and a normal small intestinal mucosa (Marsh grade 0–1) [34]. As will be discussed below, these patients may also present with anemia or ID. Subclinical CD has no signs or symptoms sufficient to trigger CD testing in routine practice. However, an unsuspected ID, without anemia, might be discovered in this subgroup if intentionally sought. Finally, refractory CD (RCD) consists of persistent or recurrent malabsorptive symptoms and signs with villous atrophy (VA) despite a strict GFD for more than 12 months. Again, anemia or ID is part of the spectrum of clinical manifestations in this subgroup”.

The prevalence of CD among patients with IDA has varied over time and possibly differs according to the geographical area studied [35]. Thus, the pooled global prevalence of CD is 1.4% based on positive results from tests for anti-tissue transglutaminase and/or anti-endomysial antibodies (called seroprevalence). However, the pooled global prevalence of biopsy-confirmed celiac disease is 0.7% (0.4% in South America, 0.5% in Africa and North America, 0.6% in Asia, and 0.8% in Europe and Oceania). The prevalence is higher in female vs male individuals (0.6% vs. 0.4%; *p* < 0.001) and is significantly greater in children than adults (0.9% vs. 0.5%; *p* < 0.001) [35]. Twelve studies assessed the prevalence of CD among patients who were evaluated for anemia [36,37,38,39,40,41,42,43,44,45,46,47,48]. In all of these, ID anemia (IDA) was the primary focus of the study or made up the cause of anemia in most of the study patients.

The prevalence of CD among patients suffering from gastro-intestinal symptoms ranged from 10.3% to 15% [36,41,42]. One small study assessed the prevalence of CD in a group of patients who had IDA but no identified gastrointestinal source [40]. In this study, the prevalence of CD by antigliadin antibody (AGA) and confirmed by endomysial antibody (EMA) was 30% [40].

In another study, the investigators assessed the prevalence of CD in premenopausal women with IDA [46]. The overall prevalence of CD in this population was found to be 12.9% by tissue transglutaminase (tTG) and 8.5% after biopsy examination confirmation. Of interest, CD was found in one of 22 (4.5%) women with hypermenorrhea and 4 of 18 (22%) women with normal menstrual flow.

Finally, four studies assessed the prevalence of CD in asymptomatic IDA patients by serology [38,40,44,45]. The prevalence of CD in this group ranged from 2.3% to 5.0%. Another three studies assessed the prevalence of CD by biopsy examination in asymptomatic IDA patients, finding it to be between 2.9% and 6% [37,42,47].

The data provided by this systematic review were biased by the criteria used for the diagnosis of CD, based in many cases on serological findings without histological confirmation [35]. Mahadev et al. performed a systematic review to determine the prevalence of biopsy verified CD in patients with IDA [48]. This systematic review consisted of examining manuscripts published in PubMed Medline or EMBASE in July 2017 for the term “celiac disease” combined with “anemia or iron-deficiency”. The authors identified 18 studies comprising 2998 patients with IDA for inclusion in the analysis. Studies originated from the United Kingdom, United States, Italy, Turkey, Iran, and Israel. Using a weighted pooled analysis, they demonstrated a prevalence of biopsy-confirmed CD of 3.2% (95% CI, 2.6–3.9%) in patients with IDA, although heterogeneity was high (I2 = 67.7%). In the eight studies fulfilling all the quality criteria established by the authors, the pooled prevalence of CD was 5.5% [48].

The CD prevalence in IDA was not influenced by the proportion of females, the average age, or the baseline prevalence of CD in the populations studied. This is notable given that IDA is more common in certain subgroups, such as premenopausal women. These findings suggest that IDA is an important risk factor for CD irrespective of patient demographics, and that endoscopic small bowel biopsy should be a part of the diagnostic workup for the condition, even in patients in which other etiologies may be suspected.

The prevalence of CD has also been investigated in patients with ID without anemia. Abdalla et al. investigated a cohort of 2105 females aged 6 years or older, obtained from the NHANES database, a nationally representative health survey conducted from 2009 to 2010 [49]. ID was defined as serum ferritin level <20 ng/mL and considered positive for CD when subjects were tested positive for both immunoglobulin A (IgA)-tTGA tissue transglutaminase and IgA-EMA. Subjects were divided into two groups (ID and non-ID). Among the sample of 2105 subjects, 569 had ID and 1536 did not have ID. Five people were identified as having CD among the ID group, as were two people in the non-ID group. After adjusting for selected covariates, the prevalence of CD was higher in female subjects with ID with OR of 12.5 (95% CI 1.74–90). These results indicate that CD is more common in patients with ID, which is in line with many different studies conducted in Europe [50,51] and Asia [52,53,54,55].

### 5.2. Children

Some authors have provided data on the prevalence of CD in children suffering from IDA. Narang et al. conducted a cross-sectional analytical study among children aged one to 12 years of age with moderate-to-severe iron IDA and control children without anemia [56]. All children with positive celiac serology underwent upper gastrointestinal endoscopy and duodenal biopsy, and a biopsy finding of Marsh grade 3 was considered positive for CD. Among a total of 152 anemic children and 152 controls with mean (SD) hemoglobin of 7.7 (1.8) and 12.2 (0.74) g/dL, respectively, 16 (10.5%) cases and 3 (2%) control patients had positive serology for CD (OR (95% CI) 5.33 (1.52–18.67), *p* = 0.007). CD was histologically confirmed in 4% of children presenting with moderate-to-severe anemia.

Of great interest is the study conducted by Shahriari et al., in which 184 children, including 92 IDA patients who responded to treatment using iron supplements, 45 non-responding iron deficient patients, and 47 healthy individuals, with the maximum age of 18 years, participated in serologic screening (with anti-TTG antibody and anti-endomysial antibody) for CD. Patients with at least one positive serology test underwent multiple mucosal biopsy from the bulb and duodenum. Interestingly, the frequency of positive serologic tests in the group with IDA resistant to treatment was prominently higher than that in the other two groups (*p* < 0.001). Among the patients with a positive serologic celiac test who underwent endoscopy and biopsy, no histologic evidence of CD was observed [57]. They were diagnosed as potentially having CD, patients with normal small intestinal mucosa who are at increased risk of developing CD as indicated by positive CD serology [58]. This suggests that even among patients with positive celiac serology and the absence of histological lesions (“potential celiac”), ID or IDA may still be present. This observation is shared by our own clinical practice.

### 5.3. Index of Suspicion for the Diagnosis of CD in Patients with IDA

Remarkably, despite the information provided in the literature, the index of suspicion for the diagnosis of CD among patients with ID or IDA is surprisingly low. Spencer et al. electronically distributed a survey to primary care physicians (PCPs) who are members of the American College of Physicians. Respondents were asked whether they would test for CD (serologic testing, referral for esophagogastroduodenoscopy (EGD], or referral to GI) in hypothetical patients with new IDA. Testing for CD varied significantly according to patient characteristics but, globally, PCPs are under-testing for CD in patients with IDA, regardless of age, gender, race, or post-menopausal status. In addition, most PCPs surveyed reported that they do not strictly adhere to established guidelines regarding a confirmatory duodenal biopsy in a patient with positive serology for CD [59].

This low awareness has also been reported among hematologists. Smukalla et al. surveyed hematologists to determine rates of CD screening in patients with IDA. The survey was e-mailed to members of the American Society of Hematology. There were 385 complete responses from 4551 e-mails. The percentage of hematologists who would indicate serology in patients with iron IDA was low, ranging from 11 to 18% compared to those who would request colonoscopy and/or gastroscopy for the study of the cause of anemia. Physicians who had recently finished their fellowship and those who saw a high volume of patients with IDA (especially if they were pediatric patients) were more likely to screen for CD [60]. The underdiagnosis of CD has serious health implications for affected individuals, and patients with newly diagnosed IDA present an opportunity for accurate diagnosis that should not be overlooked.

### 5.4. Prevalence of Celiac Disease among Patients with Anemia of Obscure Origin

Patients with IDA of unknown etiology are frequently referred to a gastroenterologist because, in most cases, the condition has a gastrointestinal origin. On the other hand, it is well known that only a minority of CD patients present with classical malabsorption symptoms of diarrhea and weight loss, whereas most patients have subclinical or silent forms in which IDA can be the sole presentation [13]. Zamani et al. investigated the prevalence of CD in a large group of patients with IDA of obscure origin. [55]. Of the 4120 IDA patients referred to a hematology department, 206 (95 male) patients were found to have IDA of obscure origin after an extensive evaluation of the gastrointestinal tract. Out of a total of 206 patients (14.6%), 30 had gluten-sensitive enteropathy (GSE) based on a positive serological test and abnormal duodenal histology. A gluten-free diet (GFD) was advised for all the GSE patients. Some results of this research deserve to be highlighted:The average duration of anemia was 3.6 +/− 1.4 years.Most of the GSE patients (73.3%) did not report any gastrointestinal symptoms. Consequently, physicians may fail to consider GSE as a cause of IDA when gastrointestinal symptoms are absent or nonspecific.These patients had been treated with oral iron for a mean duration of 1.9 years. Anemia improved in only eight patients (26.8%) treated with oral iron supplementation before GSE diagnosis.In GSE patients, the hemoglobin level was inversely correlated with the severity of the histological injury. Patients with Marsh 3 lesions had the most severe anemia, consistent with the role of impaired intestinal absorption in the pathogenesis of IDA. Many authors consider the presence of villous atrophy (e.g., Marsh 3) as one of the major criteria for diagnosing CD [61,62]. To avoid this controversy in the definition of CD, the authors used the term “gluten sensitive enteropathy” rather than CD to describe patients with any degree of intestinal damage together with positive serologic tests.In this study, the authors showed a significant objective improvement in hemoglobin levels with GFD alone in patients with positive serology but no villous atrophy (e.g., Marsh 1 or 2). This would be an important point concerning the route of iron administration (oral versus intravenous) in patients with or without villous atrophy (see below). Furthermore, GFD could improve anemia in IDA patients who have positive tTGA/EMA and mild duodenal lesions without villous atrophy.

Figure 2 shows the reported prevalence of CD in patients investigated for ID anemia in different clinical settings [35,36,37,38,40,42,44,45,46,47,49,51,56].

## 6. Prevalence of Anemia in CD

### 6.1. Comprehensive Overview

Indeed, CD is the disease of a thousand faces. Thus, a patient can be severely anemic and not have osteoporosis and vice versa without us truly knowing the reason for this phenomenon. Depending on the study in question, the prevalence of anemia in newly diagnosed patients has varied from 12% to up to 85% [38,63,64,65,66,67,68].

Saukkonen et al. compared a variety of clinical, serological, and histologic variables between newly diagnosed celiac patients presenting with and without anemia [68]. In this study, 23% of the patients had anemia at CD diagnosis. The anemia group had lower hemoglobin at CD diagnosis (women, 118 vs. 132 g/L, *p* < 0.001: men, 120 vs. 147 g/L, *p* = 0.001). Some findings that should be highlighted in this research are as follows:Celiac patients with anemia as a prominent symptom showed signs of more severe disease than those presenting with diarrhea, a finding that has also been reported by other authors [69].Anemic celiac patients have a longer duration of symptoms and a more severe serological and histologic presentation at diagnosis, as described by Shing et al. [66].Finally, anemic patients also showed a slower histologic response, including a worse recovery in the villus/crypt ratio and a significantly lower decrease in IELs count and in the density of ɣδ+ IELs.

In conclusion, this study demonstrated that CD patients presenting with anemia at diagnosis have more advanced disease and a slower dietary response than those without anemia. This observation has been supported by numerous studies as reported in an excellent review on the extraintestinal manifestations of CD, highlighting that when anemia is the main reason for presenting with the disease, they have higher anti-transglutaminase levels, lower serum cholesterol, and higher degrees of villous atrophy when compared to those presenting with diarrhea alone [70,71].

### 6.2. Anemia Outcomes in Celiac Patients after Introduction of a Gluten-Free Diet

Recovery from anemia usually occurs within one year after the commencement of a strict FGD, in most cases even without additional iron supplementation [72]. However, some patients with CD continue to have IDA despite a careful gluten-free diet (GFD). Studies on the effect of GDD on recovery from IDA are scarce [68,72,73,74]. Annibale et al. evaluated a series of 26 adult patients (24 women, 2 men; 13.7%) with a biopsy-confirmed CD diagnosis. At 12-month control, all but one patient (94.4%) recovered from anemia and 50% from ID. A significant inverse correlation (r = −0.7141, *p* = 0.0003) between an increase in Hb concentrations and a decrease in individual histological scores of duodenitis was observed, suggesting that the recovery from anemia occurs in parallel with the normalization of histological alterations of the intestinal mucosa, without iron supply. Note that the recovery of the iron stores occurred in only 50% of cases (mostly women of childbearing age). In another study, a strict gluten-free diet led to an increase in serum iron, resolution of anemia, and restitution of normal mucosal morphology in some celiac patients with severe anemia who had not responded to oral iron replacement [75].

Sansotta et al. conducted a retrospective chart review of patients contained in a registry of children (<18 years of age) and adults (>18 years of age) with CD followed at the University of Chicago between 2002 and 2015 [73]. Out of a total of 554 cases (227 children) with CD, 48% of adults and 8% of children had IDA at the time of diagnosis. All of the patients were instructed to start a strict GFD with the aid of a dietitian following their diagnosis. At the end of the follow-up period (an average of 3.4 years for children and 3.0 years for adults), 85% of adults and almost 100% of children had hemoglobin levels in the normal range.

Of particular interest are the results obtained in the study carried out at Tampere University Hospital and the University of Tampere (Finland) [68]. A total of 163 consecutive adults with confirmed CD were enrolled. The median age of the participants was 49 (range: 16 to 79) years, and 111 (68%) of them were women; 38 (23%) had anemia at CD diagnosis and 10 (6%) still after a one-year diet. Anemia was more common in women, possibly reflecting the generally higher need for iron in premenopausal women, and less common in screen-detected patients. At this point, it could be assumed that active case-finding and screening of at-risk patients has also shortened the diagnostic delay in CD and, thus, further reduced the risk of anemia at diagnosis.

Gamma-delta intraepithelial lymphocyte (ɣδ + IELs) density in the duodenal mucosa was significantly lower in the anemia group at diagnosis. This finding is of interest as there is evidence that ɣδ + IELs play an important role in mucosal repair and tumor surveillance, and that their number is reduced in patients with refractory CD [76]. It might thus be hypothesized that the low density of ɣδ + IELs contributes to the more severe presentation and increased risk of complications in anemic CD patients. In fact, these patients had more gastrointestinal symptoms, worse indicators of well-being, and higher levels of tTGA. After one year on a gluten-free diet, the mucosal villous height/crypt depth ratio was significantly lower, and all IELs except ɣδ + IELs were significantly higher in the anemia group, indicating a slower response to the GFD. These results suggest that anemia at diagnosis predisposes one to an inadequate histologic response, warranting careful follow-up for this patient subgroup.

## 7. Mechanisms that Explain the Presence of Anemia in CD

### 7.1. Micronutrient Deficiencies

CD leads to an abnormal immune response, which is followed by a chronic inflammation of the small intestinal mucosa with progressive disappearance of intestinal villi leading to a decrease in absorption of many nutrients, including iron, vitamin B12, folate, copper, and zinc [4].

Patients with CD have a predisposition to develop ID, thought to be due to the predominant site of mucosal damage—the duodenum—in CD, which is also the site of maximal iron absorption. In addition, individuals with CD are also predisposed to several other hematologic abnormalities, including vitamin B12 or folate deficiency [77,78].

Harper et al. assessed a variety of hematologic and associated nutritional parameters in a total of 405 celiac patients [64]. Approximately 20% of all patients with CD had anemia at presentation, and ~70% of CD patients had a serum ferritin below the mean for their age- and gender-matched cohort. Macrocytic anemia with concurrent vitamin B12 and/or folate deficiency was rare (<3% of all cases of anemia); however, folate deficiency was present in about 12% of the study population and vitamin B12 deficiency in about 5%. The IDA was present in 13% of the patients with partial villous atrophy and in 34% of those with a subtotal villous atrophy (*p* < 0.001), with no significant difference in the proportion of B12 and folate deficient individuals.

Similar results were reported by Berry et al. in a prospective observational study, where 103 consecutive patients with well-documented CD were included. Overall, ID was seen in 84 (81.5%) patients, followed by vitamin B12 deficiency in 14 (13.6%) and folate deficiency in 11 (10.7%) patients; 17 (16.5%) patients had anemia due to mixed nutritional deficiencies; and four (3.9%) patients had anemia of chronic disease. Again, the mean hemoglobin and median ferritin levels were significantly lower in patients with severe villous atrophy compared to those with mild atrophy [79].

The reason that some, but not all, CD patients develop ID anemia is not well understood but may be related to deficiencies in some regulatory proteins that play a critical role in iron absorption at the level of the enterocyte.

Iron enters the epithelial cell of the duodenal mucosa in ferrous form through the apical or brush border divalent metal transporter DMT1, and the efficiency of iron absorption parallels the level of DMT1 expression [80]. Thus, in the presence of ID, DMT1 increases the spanning of the entire brush border membrane instead of limiting its localization to the villus apical region [81]. Microcytic anemia caused by DMT1 mutations has also been identified in human subjects [82]. Interestingly, the DMT1 iron transporter is known to be upregulated in CD to counteract villous atrophy [83]. Taking advantage of this evidence, Tolone et al. investigated the association between an intronic DMT1 polymorphism, DMT1 IVS4+44C>A, and IDA in a cohort of 387 unrelated celiac children from southern Italy. The authors found that the DMT1 IVS4+44-AA genotype confers a fourfold risk of developing anemia, regardless of atrophy degree. The data from this study suggest, for the first time, that CD may unmask the contribution of the DMT IVS4+44C>A polymorphism to the risk of IDA [84].

In addition to the above mentioned deficiencies (i.e., iron, folic acid, and vitamin B12), at the time of diagnosis, there may be deficiencies in other vitamins and minerals, in particular copper and zinc [85]. Copper deficiency is a rare complication in CD, and its prevalence remains unknown. This deficiency can lead to anemia, thrombocytopenia, neutropenia, and peripheral neuronal involvement [86,87,88,89].

Zinc deficiency is also uncommon in celiac patients. In a study, zinc absorption did not appear below usual amounts in subjects with CD, but children with CD have impaired gut function that may affect their zinc nutritional status as shown by a smaller fractional zinc absorption compared with control patients [90]. The mechanism of zinc depletion and its possible implications are unknown [91].

### 7.2. Infection by Helicobacter pylori

*Helicobacter pylori* has been proposed to have a role in the homeostasis of iron stores, and many studies have reported that *H. pylori* infection is associated with IDA [92,93]. A systematic review and meta-analysis were conducted by Hudak et al. to examine the prevalence of depleted iron stores among patients infected with *H. pylori.* Compared to uninfected patients, *H. pylori*-infected individuals showed increased likelihood of IDA and ID [94]. Meta-analyses of seven RCTs showed increased ferritin following anti *H. pylori* eradication therapy plus iron therapy as compared with iron therapy alone. Several mechanisms have been postulated to explain this association, including elevated gastric pH due to atrophic gastritis [95,96], elevated serum hepcidin levels [97], and the presence of lymphocytic enteritis, a lesion that is shared by CD itself [98]. In a study conducted by Sapmaz, the serum hepcidin-25, iron, ferritin levels, and total iron-binding capacity were evaluated at baseline and after *H. pylori* eradication to assess whether *H. pylori* eradication plays a role in IDA related to *H. pylori* infection. There was an improvement in hemoglobin, iron, total iron-binding capacity, and ferritin values after *H. pylori* eradication in all subjects. Serum hepcidin-25 levels significantly decreased after *H. pylori* eradication (*p* < 0.001) [97]. This is important, as the presence of *H. pylori* infection upregulates serum hepcidin levels and decreases the response to oral iron therapy in children with iron deficiency anemia [99]. The presence of *H. pylori* infection could have an additive effect that may contribute to the development (or refractoriness) of anemia in CD patients. In fact, some authors have postulated a significant association between *H. pylori* infection in IDA and CD patients [100,101]. There is a low awareness among clinicians of the importance of the concomitance of both diseases (*H. pylori* infection and CD), mainly in children and adolescent groups where iron requirements are increased and eradication of the infection seems mandatory, in addition to a GFD. In conclusion, investigation and eradication of *H. pylori* should be incorporated into the IDA diagnostic workup, especially in populations where the infection is endemic [102,103].

### 7.3. Anemia of Chronic Disease

Although anemia is still a common presentation of CD, nutritional deficiencies alone do not explain this phenomenon in all cases, and CD has also been associated with other causes of anemia, such as anemia of chronic disease [4,64,104,105,106]. Systemic inflammation, subsequent to the increase in blood levels of inflammatory proteins, is a rare event in patients with CD. However, gliadin can favor the activation of mononuclear cells, located in the intestinal lamina propria mucosa, with subsequent local overproduction of proinflammatory cytokines (Figure 3). In the study conducted by Harper et al. [64], ferritin was greater than the 50th percentile in 13% of the anemic patients. Ferritin is an acute phase reactant whose serum concentration can increase in response to systemic inflammation. In fact, this subgroup showed an elevated erythrocyte sedimentation rate (ESR). The elevated ESR observed in those with high ferritin values suggests systemic inflammation without significant malabsorption. Thus, the combination of anemia associated with high serum ferritin and evidence of systemic inflammation suggests anemia of chronic disease [64]. Figure 3 illustrates the mechanisms involved in the pathogenesis of anemia of chronic disease [4,64,79,105,106].

In response to inflammation, cytokines, such as IFN-gamma, TNF-alpha, IL-1, IL-6, and IL-10, are released into circulation. These cytokines act on the liver, causing increased production of hepcidin, an acute phase reactant whose role is to inhibit the duodenal absorption of dietary iron. These cytokines also induce the expression of DMT-1, an iron transporter on macrophages, whose role is to increase iron uptake, and they simultaneously down-regulate the expression of the iron exporting protein ferroportin-1 on macrophages. The net effect is a trapping of circulating iron in the reticuloendothelial system.

### 7.4. Persistence of Anemia in Patients with CD despite Adopting a GFD

Some patients with CD persist with indicators of IDA (or ID without anemia) refractory to oral iron supplementation after adopting a GFD. Several factors may be involved in this well-documented phenomenon. First, although strict avoidance of gluten typically results in clinical and histological improvement, when a follow-up duodenal biopsy is performed to document mucosal recovery, a significant proportion of patients with CD persist with villous atrophy. This fact has recently been highlighted by Fernández-Bañares et al. in the CADER study, which was designed to evaluate villous atrophy persistence after two years on a GFD in de novo adult patients with CD with strict control of gluten exposure [107]. Seventy-six patients completed the study (36.5 ± 1.6 years, 73% women). The rate of persistent villous atrophy after two years was high (53%) in adult patients with CD on an intentionally strict GFD, despite rigorous prospective dietary monitoring. Two-thirds of participants (69%) had detectable gluten immunogenic peptides in the fecal sample (f-GIPs > 0.08 mg/g) during the study period, without significant differences between patients who achieved recovery and those with persistent villous atrophy. The authors conclude that low-level ongoing inadvertent gluten exposures could be a contributing factor to persistent villous atrophy in highly sensitive patients [107].

Secondly, anemia that is refractory to oral iron supplementation is also seen in CD patients, despite the recovery of duodenal mucosa after adopting a GFD. Several authors have reported the presence of ultrastructural and/or molecular alterations of enterocytes, such as disrupted and decreased glycocalyx, and irregular or absent microvilli after adopting a GFD [4]. Such findings were only visible when biopsies were investigated by medium- or high-power scanning electron microscopy, whereas low-power scanning electron microscopy did not detect these lesions. Such changes would seem to be involved in the persistence of IDA in this subgroup [108,109,110]. Ultrastructural and molecular alterations of enterocytes would also appear to be responsible for IDA in patients with non-celiac gluten sensitivity (NCGS), where IDA is present in up to 20% of cases [111,112]. This hypothesis is consistent with the findings observed by the group of Sbarbati et al. The authors revealed alterations of the enterocyte brush border with a significant reduction in the height of microvilli in four out of seven patients with gluten sensitivity, alterations that were not detected by conventional light microscopy [113].

### 7.5. Blood Loss Due to Inflammatory Lesions

Blood loss due to inflammatory lesions of intestinal mucosa may contribute to IDA in celiac patients [4,106]. Such is the case for patients with concomitant inflammatory bowel disease (IBD) [114], as well as those who have developed refractory celiac disease (RCD) and/or associated complications, including enteropathy associated with T-cell lymphoma (EATL), adenocarcinoma, jejunoileitis, or B-cell lymphoma [115,116]. In this regard, small bowel capsule endoscopy (SBCE) has an established role in the identification and management of IBD and RCD lesions and for the detection of complications [117,118,119,120,121]. The sprue collagen manifests itself in the form of refractoriness, and its occasional association with EATL has also been described [122].

### 7.6. Aplastic Anemia

Various cases of aplastic anemia associated with CD have been described in the literature, both in pediatric age and in adulthood [123,124,125,126,127,128]. Pancytopenia is the most common feature, and the clinician should have a high index of clinical suspicion for this entity in the presence of pancytopenia that complicates the evolution of CD. The diagnosis is usually achieved by bone marrow biopsy. Based on the cases reported to date, it seems that the GFD is not enough to improve pancytopenia; therefore, most patients require other treatments.

Figure 4 illustrates the various mechanisms underlying the pathogenesis of anemia in the CD patient.

## 8. Management of Anemia and Iron Deficiency in Different CD Settings (Algorithms)

### 8.1. At the Time of CD Disease Diagnosis

A variable proportion of CD patients (children or adults) do not have anemia at diagnosis. However, some of them have indicators of ID and are symptomatic (e.g., asthenia, fatigue, and poor exercise tolerance). Therefore, the clinician should order a battery of laboratory tests to identify any signs suggestive of iron deposition depletion and to act accordingly. ID is diagnosed by the presence of low serum iron levels (less than 50 µg/dL) and high serum transferrin level of IBC > 350 mg/dL. Another highly sensitive index for diagnosing IDA is the saturation index of transferrin, which is less than 10–16%. Low ferritin levels represent an early and highly specific indicator of iron deficiency. However, the international criteria for defining depleted iron deposits vary with age: <12 µg/L for children under five years of age and equal to 15–20 µg/L for those over 5 fiveyears of age and adults [106]. These levels should be adjusted upwards in the presence of inflammation or infection, so that the cut-off for the diagnosis of IDA rises to 30–50 µg/L in this context. Table 1 shows other parameters for IDA diagnosis.

### 8.2. Allogeneic Red Blood Cell Transfusion (When, How, and to Whom?)

Acute post-hemorrhagic anemia is a rare event in the celiac patient, except in cases of complicated CD, especially if patients are receiving anticoagulants [129,130,131,132]. The indications for red blood cell (RBC) transfusion in CD patients do not differ from those established for the general population, and all of these patients should benefit from a patient blood management (PBM) program that seeks to minimize blood loss, optimize hematopoiesis (mainly with iron replacement therapy), maximize tolerance of anemia, and avoid unnecessary transfusions [133,134,135]. Table 2 shows the indications and hemoglobin thresholds for RBC transfusion in the adult with acute or chronic anemia due to gastrointestinal bleeding or malabsorption [136].

RBC transfusion is not usually necessary in CD with chronic IDA, except in cases where a source of chronic gastrointestinal bleeding (e.g., angiodysplasias) is identified, especially in elderly patients, where CD may also occur and coexist with some co-morbidities, such as coronary insufficiency, heart failure, chronic obstructive pulmonary disease (COPD), or renal failure. All of these can exacerbate the effects of tissue hypoxia associated with the loss of red cell mass and may precipitate organ failure. This reflection is important since as the prevalence of CD increases, a greater proportion of new diagnoses are being made in individuals over 60 years of age [137,138,139,140]. The atypical (non-classics) patterns of clinical presentation in this age group and the lower sensitivity and specificity of serological tests in the aged population can sometimes cause a delay in diagnosis [140]. It is unknown why the gastrointestinal pattern (GI) of presentation is less common in the elderly than in younger adults; however, in the aged population, deficiency of micronutrients may often represent the only symptom at presentation [141]. Thus, up to 80% of elderly patients with CD in several countries presented with anemia, mainly due to iron deficiency [52,137,138,139,142]. The clinician must be vigilant since the etiology of anemia in elderly patients includes factors other than ID, such as vitamin B12, folate deficiency and inflammation itself, and some hematological parameters; for example, MCV and ferritin levels may be equivocal [139,141].

### 8.3. Replenishment of Iron Storage

Iron is an important micronutrient, and CD constitutes one of the groups at the highest risk of ID. ID during the first year of life occurs at a time of rapid neural development and when morphological, biochemical, and bioenergetic alterations may all influence future functioning [22,23,24]. The brain is the most vulnerable organ during critical periods of development [19]. Iron is present in the brain from very early in life, when it participates in the neural myelination processes [20], learning, and interacting behaviors, and iron is needed by enzymes involved in the synthesis of serotonin and dopamine neurotransmitters [21]. In adults, IDA results in fatigue and diminished muscular oxygenation, which may affect muscle strength and quality and, subsequently, physical performance. In both populations (children and adults), iron deficiency leads to increased vulnerability to infections, especially of the respiratory tract. Consequently, it is important to replenish iron stores quickly, safely, and effectively in any patient with a gastrointestinal source of ID, either through blood loss or malabsorption.

### 8.4. How to Proceed with Iron Replenishment: When, How, and to Whom?

GFD favors the improvement of intestinal atrophy but also induces a reduction in inflammation with subsequent progressive correction of anemia. The mechanism is therefore twofold: increased iron absorption and reduced effects of various inflammatory mediators on iron homeostasis and erythropoiesis. The recovery from anemia occurs in parallel with the normalization of histological alterations of the intestinal mucosa, without iron supply, although the recovery of the iron stores occurs in only 50% of cases. At this point, the indications for iron replacement are determined by the impact of ID on symptoms, HRQL, reduced work productivity, and the deleterious effects of anemia on comorbidities, especially in older patients.

### 8.5. Oral or Intravenous Iron?

The decision to replenish iron stores by oral or intravenous iron administration is controversial [143,144,145] and depends primarily on the severity of symptoms, the tolerance and efficiency of oral iron, and those factors that predict a poor response to oral iron (e.g., severity of histological lesion; poor adherence to GFD; or blood loss due to mucosal lesions, such as concomitant IBD, jejunoileitis, or malignancy). Figure 5 is an algorithm proposed by the authors for decision making. This proposal should be validated by well-designed studies comparing the efficiency and safety of both replenishment routes in the different settings indicated and their cost-effectiveness.

### 8.6. Oral Iron Considerations

GFD alone may improve mild forms of IDA in patients with CD [68,72,73,74]. In fact, GFD is the primary means of preventing anemia in CD patients after diagnosis. However, recovery may be slow [6,83,146,147], and the administration of iron may accelerate the replenishment of iron stores in the body and thus the resolution of ID-dependent symptoms. This strategy can be useful especially in those patients with mild forms of enteropathy (Marsh 1-3a), especially if adherence to the GFD is not good. There are many barriers to following a GFD because gluten is present in many foods, and the cross-contamination is always a cause for concern. Dietary counseling that provides adequate and thorough information to the patients and their families regarding this disease and the need for lifelong adherence to a GFD is necessary. During follow-up, it is important to investigate for a micronutrient deficiency, such as of iron (a complete blood count plus serum ferritin), calcium, folic acid, vitamin B-6, and vitamin B-12 [148]. In contrast, replacement therapy with oral iron formulations is often ineffective and poorly tolerated in patients with more advanced forms of enteropathy (Marsh 3b-3c), because unabsorbed iron impregnates and irritates the duodenal mucosa and is the cause of numerous adverse effects [149]. Table 3 shows some considerations of interest in relation to oral iron replacement [150]. By way of summary:Ferrous sulphate (FS) is the most undertaken therapy for oral iron replacement [106,151].In children, the recommended iron dose is 2–6 mg/kg/day in terms of elemental iron. In adolescents and adults, it is 100–200 mg daily. Sometimes, these high doses of oral iron cause a paradoxical decrease in iron absorption due to factors such as elevated plasma hepcidin levels [152,153]. In our practice, formulations that provide 40–80 mg of elemental iron, when administered once (80 mg) or twice (40 mg/12 h) daily, are equally effective and better tolerated.Toxicity associated with oral iron is higher in elderly patients, and such patients should be treated with lower doses. In fact, doses of 15, 50, or 150 mg of elemental iron may be equally effective in raising hemoglobin and ferritin levels, while adverse effects are significantly less common with lower doses [154].Strategies for reducing side effects and improving tolerability include:o Limiting the dose (≤80–100 mg of elemental iron per day).o Dividing the total dose and taking it in two daily doses or increasing the time between doses (e.g., every two days) [155].o Taking iron after dinner (reduces absorption but improves tolerance).o Changing the formulation (e.g., from ferrous sulphate to ferrous gluconate) or presentation used (e.g., from tablets to oral solution, which makes it easier to titrate doses).o Some proposed solutions to improve oral iron absorption in CD include the use of probiotics (*Lactobacillus plantarum 299v* and *Bifidobacterium lactis HN019*) [156,157] or prebiotics (oligofructose enriched inulin) [158], as well as the use of ferrous bisglycinate chelate (FBC), or the most recent Feralgine^®^, a compound of FBC and alginic acid that has recently been developed to improve the bioavailability and tolerability profile. Feralgine^®^ and FBC are effective at a dosage of 30–40% compared to FS. Several studies have demonstrated the efficacy and safety of FBC in the treatment of IDA in both adults and children, without showing side effects [159,160,161,162]. In addition, recent studies conducted in adult celiac patients confirmed the good level of absorption and tolerance of Feralgine^®^ in patients with anemia as well as in non-celiac subjects and in those with onset CD [163,164,165].o Another alternative aimed at reducing the risk of adverse effects associated with iron sulphate is sucrosomial iron (SI). SI a preparation of ferric pyrophosphate covered by a phospholipids and sucrester membrane, can be absorbed across intestinal epithelium by an alternative route, non-mediated by the DMT-1 carrier [166], which may contribute to the reduction of side effects and the prevention of iron instability in the gastrointestinal tract. A study evaluated the efficacy and safety of a new SI formulation (30 mg of iron/day) versus iron sulfate (105 mg of iron/day), in patients with CD. After a follow-up of 90 days both groups showed an increase in Hb levels compared to baseline (+10.1% and +16.2% for sucrosomial and sulfate groups, respectively), and a significant improvement in all iron parameters, with no statistical difference between the two groups. However, patients treated with SI reported a lower severity of abdominal symptoms, such as abdominal and epigastric pain, abdominal bloating, and constipation, and a higher increase in general well-being (+33% vs. +21%) compared to the iron sulfate group [167]. Therefore, SI can be effective in providing iron supplementation in difficult-to-treat populations, such as patients with CD, IDA, and known intolerance to iron sulfate.Response to oral iron therapy can be considered satisfactory when an increase in hemoglobin levels of at least 2 g/dL is observed within 3–4 weeks, which is also associated with an improvement in physical well-being and anemia-dependent signs and symptoms, including depapillation of the sides of the tongue, which is a good indicator of recovery. For patients with persistent anemia or ID and doubts about correct adherence to the GFD, it may be important to investigate the presence of gluten immunogenic peptides (GIPs) in fecal or urine samples, as these are present in a significant proportion of patients who declare a correct adherence to the diet. This policy may avoid unnecessary biopsies or limit them to cases where symptoms persist despite good nutritional advice and repeatedly negative GIP results [168].If oral iron is not tolerated, or not absorbed due to intestinal inflammation, then intravenous iron should be given.

### 8.7. Intravenous Iron Replacement Therapy

The scenarios in which parenteral iron replacement may be indicated in patients with CD and IDA or ID are diverse and are reflected in Figure 5. These can be grouped into two categories: (1) When there is a clinical need to deliver iron rapidly. Such is the case of patients with severe anemia, often of multifactorial etiology, or poorly tolerated due to the presence of comorbidities (especially in the elderly). In these cases, RBC replenishment is not sufficient to restore the depleted iron stores. (2) When oral iron preparations are not tolerated or not absorbed due to intestinal inflammation, which is highly likely in patients with poorly controlled enteropathy (total or subtotal villous atrophy), poor adherence to the GFD, or concomitant inflammatory status [169]. A third scenario to consider is that of patients with a marked deterioration in HRQL (including fatigue, weakness, poor exercise tolerance, and lack of concentration) or evident risk of neural development disorders (children’s) [68,143,144,145,170,171]. HRQL instruments provide a means of exploring patient perceptions of the effects of IDA/ID on daily living and thus provide additional information that cannot be directly extrapolated from clinical measures. Regarding this point, it is not unusual for the deterioration in HRQL to be recognized only when the iron deposits have been restored in a rapid, safe, and effective manner [172,173,174].

### 8.8. Iron Formulations for Intravenous Use

Numerous various iron formulations for intravenous use have been developed in recent years. Their efficacy in the management of IDA is greater than the possible adverse events, and, in fact, the rates of mild reactions are ~1 in 200 and those of major reactions are ~1 in 200,000 or more [106]. Table 4 shows the characteristics of the different iron formulations available and Table 5 the advantages and limitations of oral versus intravenous iron. Figure 6 shows a simple algorithm for calculating the dose of iron to be administered based on the patient’s weight and hemoglobin level. This pragmatic approach is easier to apply than the classical Ganzoni formula, which, in some contexts, may underestimate the real iron needs.
Ganzoni Equation for Iron Deficiency Anemia
Total, iron deficit (mg) =
Body weight (kg) × (Target Hb − Actual Hb) [g/dL] × 2.4
+
iron stores (mg)

Target Hb: 13 g/dL for a body weight of less than 35 kg and 15 g/dL for a body weight of more than 35 kg.

Factor 2.4 = 0.0034 × 0.07 × 10,000, where:▪0.0034: iron content of hemoglobin (0.34%).▪0.07: blood volume 70 mL/kg of body weight = 7% of body weight.▪10,000: conversion factor 1 g/dL = 1000 mg/L.▪iron stores (mg): 500 mg if the body weight is greater than 35 kg or 15 mg/kg if the body weight is less than 35 kg.

### 8.9. Intravenous Iron in Children

ID with or without anemia is a common complication of pediatric CD, causing significant morbidity. Despite this, ID remains prevalent and undertreated, related in part to questions surrounding optimal formulation and route of administration. In addition, its application in daily management has been overlooked, partly due to the fear of possible adverse events related to historical anaphylactic reactions associated with iron dextran formulations [101,174]. However, in the case of severe anemia, it might be taken into consideration in order to rapidly correct the hematological picture. Some studies have shown that intravenous iron administration is safe and effective in children with various diseases leading to IDA. Carman et al. conducted a study on a total of 101 pediatric patients 6–18 years of age with IBD and iron deficiency (ID) or iron deficiency anemia (IDA). Patients received ferric carboxymaltose (FCM), a recent formulation of intravenous iron, allowing higher doses and rapid infusion times. Following FCM infusion as a single dose of 15 mg/kg up to 1000 mg over 15–20 min, 64% of patients with IDA had resolution of anemia, with 81% showing resolution for ID without anemia [175]. Similar data have been reported by other authors [176,177].

CD (and, for that matter, any other disease causing villous atrophy) is a classic scenario where the results of oral iron replacement are poor due impaired absorption inherent to mucosal injury. These children could benefit from a strategy focused on intravenous iron administration under certain conditions (see below). In a retrospective cohort study, a total of 116 IV iron carboxymaltose infusions were administered to 72 patients with IDA refractory to oral iron and were shown to be safe and highly effective in a small yet diverse population of infants, children, and adolescents [178,179].

Among the available intravenous iron preparations, only iron sucrose [180,181], iron carboxymaltose [175,176,178,179,182], and low molecular weight iron dextran [183] have been studied in children, but, to date, none has a pediatric indication. A panel of experts led by Mattiello (SPOG Pediatric Hematology Working Group) has recently provided ID management recommendations based on the best available evidence in response to one of the most common challenges faced by pediatricians [184]. In accordance with these recommendations, IV iron administration can be considered a first-line strategy in patients with chronic IBD or situations with proven malabsorption and a second-line strategy after consultation with a specialist in pediatric iron metabolism (certified pediatric hematologist) (Table 6) under specific conditions:(1)Failure to achieve correction of IDA after well-conducted oral iron substitution in the setting of good adherence of at least six months of prescribed supplementation and two formulation attempts.(2)Confirmed malabsorption or chronic oral iron intolerance, including the category of children with severe neurological/neurodevelopmental impairments leading to feeding limitations.

### 8.10. Adverse Effects and Contraindications Related to the Use of Intravenous Iron

Data from studies in adults show that IV iron is contraindicated in the course of infections, in the first trimester of pregnancy, and in patients with a history of iron or of another significant (i.e., anaphylactic) drug allergy. Immediate side effects of an IV iron infusion can be nausea, vomiting, headache, flushing, myalgia, pruritus, arthralgia, and back and chest pain. Hypophosphatemia can be observed but is usually transient and asymptomatic. Possible adverse events can be easily managed if suitable measures are implemented to ensure early diagnosis and effective management of allergic reactions (Table 6 and Table 7) [150].

## 9. Summary and Conclusions

ID with or without anemia is a common complication in both children and adults with CD, causing significant morbidity and impairment of HRQL. In some cases, IDA/ID is the main or even the only clinical manifestation of the disease. Importantly, the presence of IDA or ID that does not reverse after oral iron administration should strongly raise the suspicion of CD to ensure that diagnosis is not delayed. Folate and vitamin B12 malabsorption, nutritional deficiencies, inflammation, blood loss, development of refractory CD, and concomitant *H. pylori* infection are other causes of anemia in such patients. Once IDA and ID are detected, the physician must restore iron stores to avoid the deleterious effects of red blood cell mass loss on physical, emotional, and psychological well-being. The decision to replenish iron stores by oral or intravenous iron administration is controversial and depends primarily on the severity of symptoms, its impact on the HRQL, the tolerance and efficiency of oral iron, and those factors that predict a poor response to oral iron (e.g., the severity of the histological lesion, poor adherence to GFD, and blood loss due to mucosal lesions).

Only when severe anemia is present, with hemoglobin values below 5–6 g/dL (which requires rapid correction such as in patients with cardiac dysfunction) can RBC transfusions be performed following the principles of the PBM strategy.

Finally, the high percentage of subjects with IDA who are celiac reinforces the need for screening CD in patients with IDA or ID.

## Figures and Tables

**Figure 1 nutrients-13-03437-f001:**
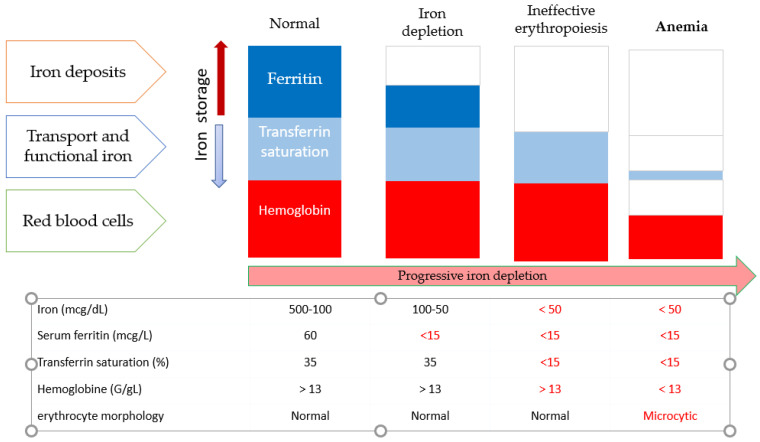
Diagnosis of iron deficiency.

**Figure 2 nutrients-13-03437-f002:**
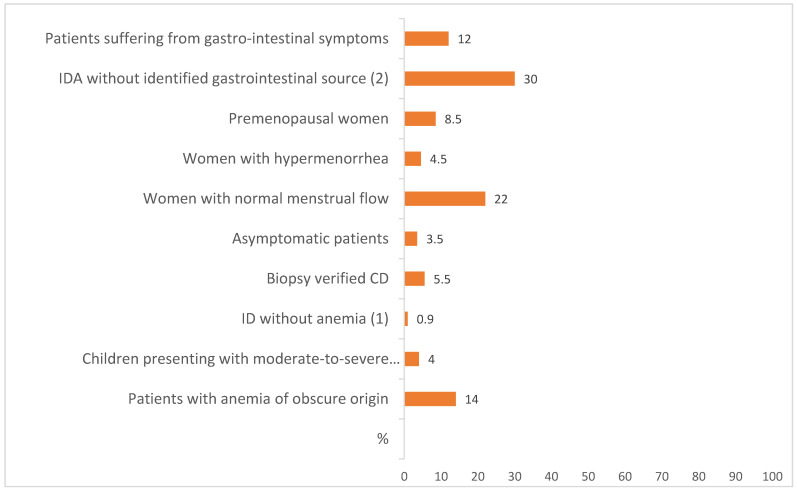
Prevalence of CD in patients with IDA in different setting. (1) OR of 12.5 (95% CI 1.74–90) (compared to the prevalence of CD controls); (2) Diagnosis celiac desease based on serology results without duodenal biopsy.

**Figure 3 nutrients-13-03437-f003:**
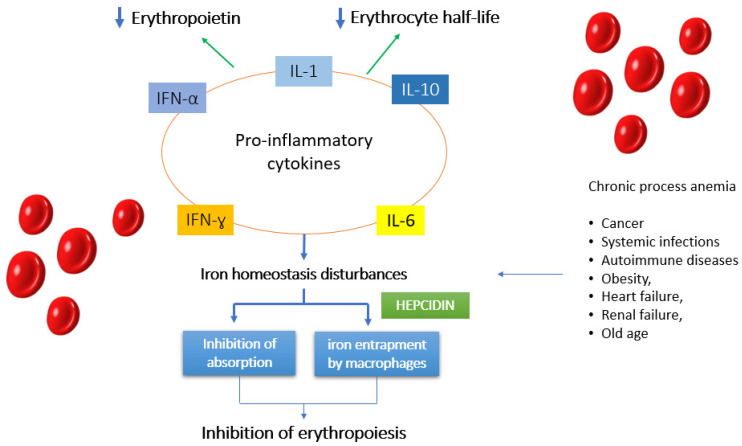
In CD there is a component of systemic inflammation which in some cases may also contribute to the pathogenesis of anaemia. This contribution is less than that due to malabsorption. The figure illustrates some mechanisms involved in the pathogenesis of anemia of chronic disease.

**Figure 4 nutrients-13-03437-f004:**
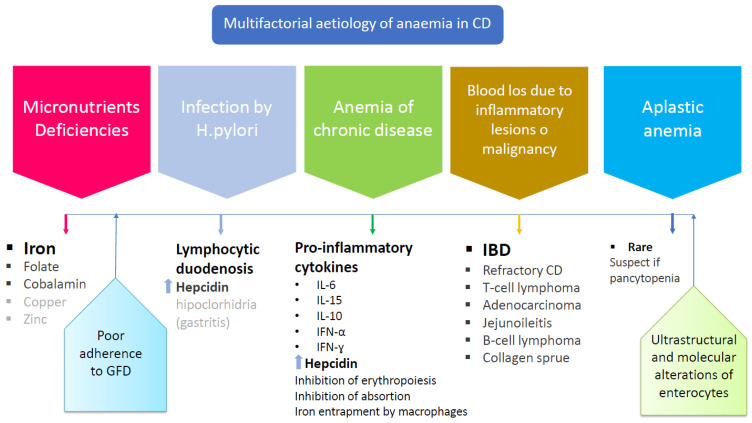
Factors influencing the development of anemia in CD. The main and most frequent cause of anemia in CD is malabsorption. The figure illustrates other components that may contribute to its pathogenesis to a lesser extent. CD “celiac disease”; GFD: “gluten free diet”; IBD: “inflammatory bowel disease”.

**Figure 5 nutrients-13-03437-f005:**
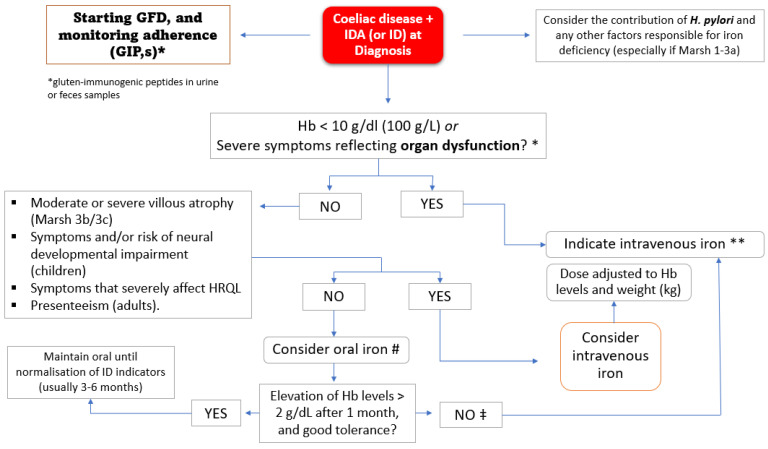
A proposed algorithm for iron replacement in patients with CD and IDA or ID. GFD: gluten-free diet; GIPs: gluten-immunogenic peptides in urine or feces samples; HRQL: health-related quality of life. # In the presence of mild symptoms, consider not giving oral iron and wait for resolution of lesions after GFD. ‡ Some causes of non-response to oral iron replacement include poor adherence to the GFD (consider the presence of GIPs in urine or stool samples as an indicator of poor adherence), slow histological response to the GFD (“slow responders”), refractory celiac, and blood loss due to mucosal lesions (Crohn’s disease, jejunoileitis, and malignancy). (*: Consider transfusion of red blood cells.; ** See Tables 5 and 6).

**Figure 6 nutrients-13-03437-f006:**
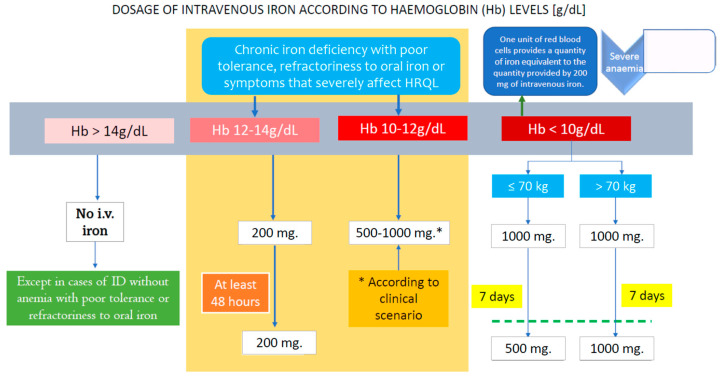
Dosage of intravenous iron according to hemoglobin (Hb) levels (g/dL) (adults). * Dosage may vary depending on the patient’s clinical condition and the physician’s clinical judgment. Hb: haemoglobin; h: hours; kg: kilograms of patient weight.

**Table 1 nutrients-13-03437-t001:** Other parameters evaluable for IDA diagnosis [106].

Parameter	Comment
✓ Reduction in Hb and hematocrit < 2 SD of normal values^*^. ✓ Reduction in MCV, MCH, and MCHC ✓ Hypochromic cells with a tendency to microcytosis ✓ Increase in RDW > 15% ✓ Reduction in CHr < 27.5 pg ✓ Increase in sTfR to a 10–14 mg/L ✓ Reduction in reticulocyte (inconstant) ✓ Increase in free erythrocyte protoporphyrin (FEP) > 10 mg/dL	* These values vary according to age, sex, elevation, smoking habit, and physiological conditions such as pregnancy.

* For WHO in adults, anemia is defined as hemoglobin < 13 g/dL in men and <12 g/dL in non-pregnant women. In children, reference values are lower and differ according to age. (MCV: mean corpuscular volume; MCH: mean hemoglobin concentration; MCHC: mean corpuscular hemoglobin concentration; RDW: red cell distribution width; CHr: reticulocyte hemoglobin concentration; sTfR: soluble transferrin receptor).

**Table 2 nutrients-13-03437-t002:** Indications and hemoglobin thresholds for RBC transfusion in the adult with acute or chronic anemia.

Hemoglobin Thresholds for RBC Transfusion # ‡	
*Acute anemia*	*Chronic anemia* ^1^
Hb < 7 g/dL In those patients without cardiovascular or pulmonary comorbidities [A] or signs of organ dysfunction [B].	Hb < 5 g/dL In those patients without cardiovascular or pulmonary comorbidities [A] or signs of organ dysfunction [B].
Hb < 8 g/dL In those patients with cardiovascular or pulmonary comorbidities [A].	Hb < 6 g/dL Only in those patients with cardiovascular or pulmonary comorbidities [A].
Hb < 9–10 g/dL In those patients with signs of organ dysfunction [B].	Hb < 7 g/dL Only in those patients with signs of organ dysfunction [B].
[A] Cardiovascular risk factors influencing the decision to transfuse RBC concentrates in patients with acute anemia: ▪ Acute coronary syndrome with active ischemia. ▪ Anginal chest pain ▪ ECG changes suggestive of ischemia ▪ Orthostatic hypotension or tachycardia that does not respond to fluid resuscitation ▪ Severe dyspnea or tachypnea at rest	[B] Signs of organ dysfunction They are indicative of severe tissue hypoxia (e.g., in cases of massive hemorrhage, where hemoglobin levels remain “elevated” due to hemoconcentration). ▪ Tachycardia ▪ Hypotension ▪ Dyspnea ▪ Angina ▪ Hypoxia
▪ Coronary artery disease, heart failure, planned surgery this admission, or coronary artery syndrome without active ischemia o Consider transfusion if Hb < 7.5 g/dL* [135].	

^1^ Iron deficiency anemia due to malabsorption or chronic fecal losses. # Transfusing RBC units one unit at a time, assessing patients after each unit (i.e., “don’t give two without review”); ‡ Each unit of RBC provides approximately 200 mg of iron, which is insufficient to replenish iron stores.

**Table 3 nutrients-13-03437-t003:** Guidance and considerations in relation to oral iron replacement.

Guidance and Considerations in Relation to Oral Iron Replacement
The dose of oral iron depends on patient age, the estimated iron deficit, how quickly it needs to be corrected, and side effects.
Absorption improves when iron is taken in a moderately acidic medium; therefore, it is recommended that iron be taken with ascorbic acid (250–300 mg) or half a glass of orange juice. Some ferric gluconate formulations contain ascorbic acid with 80 mg of elemental iron.
Some food components, such as phosphates, phytates, and tannates (which are found in coffee, tea, cocoa, and red wine), inhibit iron absorption. Other foodstuffs that impair iron absorption are cereals, dietary fiber, eggs, milk, and generally any foods with a high calcium content. Many of these items regularly form part of patients’ breakfasts. The summary of product characteristics for most oral iron products therefore recommend taking oral iron at least 1 h before or 2 h after eating. However, although the administration of oral iron together with food decreases absorption, it improves tolerance and is one of the strategies used by many doctors in the event of side effects (see above).
Iron is best absorbed as the ferrous (Fe++) salt in a mildly acidic medium. Gastric acidity is helpful and medications that reduce gastric acid (e.g., antacids, histamine receptor blockers, proton pump inhibitors) may impair iron absorption. Other medications that impair oral iron absorption are calcium supplements and certain antibiotics (quinolones and tetracyclines), and, therefore, oral iron should be taken at least 2 h before or after these medications.
Enteric-coated or sustained-release capsules are less efficient for oral absorption because iron is released too far distally in the intestinal tract (or not at all).
Gastrointestinal symptoms associated with taking oral iron are common and include metallic taste, dyspepsia, nausea, vomiting, flatulence, diarrhea, and constipation. Some patients may also be bothered by the dark green or tarry stools (they should be warned if they are to undergo a colonoscopy). As a result of this, compliance with oral iron administration may be low. The severity and impact of these effects has been demonstrated in various systematic reviews and meta-analyses of randomized studies [149,166], and they are estimated to affect 30–43% of patients, depending on the formulation used. Supplements containing smaller amounts of elemental iron are associated with less gastrointestinal toxicity, especially in elderly patients [154]. Taking iron after dinner reduces absorption but improves tolerance. The reader is referred to the recommended doses in Section 8.6 of the text.

**Table 4 nutrients-13-03437-t004:** Characteristics of the different iron formulations available ^1^.

Brand Name	Venofer^®^, Feriv^®^ (Iron Sucrose)	Ferlixit^®^, Ferrlecit^®^ (Fe-Gluconate) ^1^	CosmoFer^®^ (Iron Dextran)	Ferinject^®^ (Ferric Carboxymaltose)
Indication	Iron deficiency	Iron deficiency	Iron deficiency	Iron deficiency
Max. iron dose in one infusion	200 mg	125 mg (12.5 mg/mL) ^2,3^	20 mg/kg of body weight	1000 mg
Duration of the dose by injection	30 min	1 h	4–6 h	15 min
Max. iron dose by injection	200 mg (3 times/week)	125 mg (12.5 mg/mL) ^2,3^	200 mg (3 times/week)	1000 mg (once/week)
No. of hosp. visits for adm. 1000 mg	5	8	1 by infusion 5 by injection	1

^1^ New preparations such as Fe-isomaltoside (Monofer^®^) and ferumoxytol (Ferraheme^®^) are currently being studied. ^2^ Product used to treat iron deficiency anemia in adults and children six years and older with chronic kidney disease receiving hemodialysis and supplemental epoetin therapy. ^3^ Ferrlecit is available in generic form. Data from Ferrlecit postmarketing spontaneous reports indicate that individual doses exceeding 125 mg may be associated with a higher incidence and/or severity of adverse events.

**Table 5 nutrients-13-03437-t005:** Comparative of oral and intravenous iron in the management of IDA.

Oral vs. Intravenous Iron Replacement	Advantages	Limitations
Oral iron	▪ Effective in many patients at right dose ▪ Low cost ▪ Low serious adverse effects	▪ Low efficiency in malabsorptive states ▪ Poor gastrointestinal tolerance ▪ Unsuitable in cases of continuous occult gastrointestinal bleeding (e.g., in the presence of concomitant mucosal lesions) ▪ It may take a long time to replenish iron stores
Intravenous iron	▪ Effective in most cases ▪ Faster correction of anemia ▪ High doses administered in a single infusion ▪ Adherence is guaranteed ▪ No gastrointestinal side effects	▪ Intravenous infusion requires monitoring ▪ Infusion-related reactions and allergies have been reported ▪ Special equipment and trained staff are required to treat potential infusion-related reactions ▪ High initial cost

**Table 6 nutrients-13-03437-t006:** Recommendations for the administration of intravenous iron.

Recommendations for the Administration of Intravenous Iron
▪ Intravenous iron preparations should only be used at centers that have immediate access to emergency treatments for hypersensitivity reactions. ▪ The administration of a test dose is not recommended since cases of allergic reactions have been reported in patients who had previously tolerated the product well. The patient should be monitored for at least 30 min after administration. ▪ Intravenous iron preparations are contraindicated in patients who are hypersensitive to any of the components of the medication and should not be used in patients who have suffered severe hypersensitivity reactions to a different preparation. ▪ Special care should be taken in patients with known allergies to other medications or with immune or inflammatory diseases, such as patients with a history of asthma or eczema or atopic patients. ▪ These preparations should only be used during pregnancy if they are clearly necessary, and their use should be restricted during the second and third trimesters to protect the fetus from potential adverse effects as much as possible. ▪ Finally, it is important to remember to report all suspected adverse reactions to the corresponding Autonomous Pharmacovigilance Centre.

**Table 7 nutrients-13-03437-t007:** Actions to be taken in the event of adverse effects.

Actions to Be Taken in the Event of Adverse Effects
MILD ▪ Discontinue the infusion until all symptoms disappear ▪ Restart the infusion at a slower rate (slower infusion rate) MODERATE ▪ Discontinue the infusion ▪ Administer 1 mg/kg of intravenous methylprednisolone ▪ Monitor the patient for 4 h or until all symptoms disappear SEVERE ▪ Administer 1000 mL of saline solution, oxygen (if required), 0.5 mg of intramuscular adrenaline, and 200 mg of intravenous hydrocortisone and admit to hospital if necessary.

## Data Availability

Not applicable.

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
