# Peer review of "Iron Deficiency in Celiac Disease: Prevalence, Health Impact, and Clinical Management"

_nutrients, 2021, doi:10.3390/nu13103437_

Round 1
Reviewer 1 Report
Dear Authors,
This manuscript is well written, but introduction is excessively long and could benefit from same rework to shorten it, particularly in the introduction section. Perhaps, the whole physiology part can be a dedicated topic for another paper given the clear effort you have made.
Please use the complete word instead of abbreviation the first time that you use the word in the text.
paragraph 5: For celiac disease definition and classification you have to use Oslo definition. In this paragraph you need to explain potential celiac disease and its follow-up ( PMID 31772571 30978358 22704250). ten lines for the whole CD definition, classification is not enough in my opinion. Please consider discussing the role of gluten free diet in this paragraph
Data of CD prevalence in the word are old, please up-date this section (PMID: 29551598)
Paragraph 8.2 would be the last option for treatment of CD anemia. Move it to the end of the manuscript
It could be useful to write a paragraph on follow-up because in CD patients, a correction of anemia appeared if gluten free diet is well done ( PMID: 28298278 ).
It could be useful to stress that GFD is the main fact to prevent anemia in CD patients after the diagnosis, in fact, as well explained in introduction, the main cause of anemia is villous atrofia.
Could be interesting a paragraph where you explain the necessity of check the compliance in CD patients when anemia continues and if necessary could be useful to do other biopsies.
Figure 3 is counfounding. Anemia of CD patients is not an anemia from chronic condition, but it is due to malabsorption.
Tables are well written and proposed algorithms are easy to understand.
Author Response
Dear reviewer,
Thank you very much for your comments and advice.
"Please see the attachment."

Reviewer 2 Report
Dear Sir
I've carefully read the manuscript entitled "Iron deficiency in celiac disease: prevalence, health impact, and clinical management" by Montoro-Huguet et al. The Manuscript is an extensive review on iron deficiency anemia in celiac disease. It is well written and include all the aspects of IDA and CD from diagnosis, pathigenesis and treatment.
My only advise is to discuss in the therapeutic paragraph the effect of liposomial iron and high iron diet in patients affected by celiac disease (Elli et alNutrients. 2018 Mar 9;10(3):330. doi: 10.3390/nu10030330; Elli l. Nutrients. 2020 Jul 17;12(7):2122. doi: 10.3390/nu12072122.)
Author Response
Dear reviewer,
Thank you very much for your comments and advice.

Round 2
Reviewer 1 Report
The Authors replied to all my concerns. They changed paragraph abaut celiac disease and improved the paragraph about treatment.